# ADVERSARIAL ATTACKS ON FINE-TUNED LLMs

## ABSTRACT

Large Language Models (LLMs) have greatly advanced the field of General Artificial Intelligence, yet their security vulnerabilities remain a pressing issue, particularly in fine-tuned models. Adversarial attacks in black-box settings—where model details and training data are obscured—are an emerging area of research, posing a substantial threat to private models' integrity. In this work, we uncover a new attack vector: adversaries can exploit the similarities between open-source LLMs and fine-tuned private models to transfer adversarial examples. We introduce a novel attack strategy that generates adversarial examples on open-source models and fine-tunes them to target private, black-box models. Our experiments show that these attacks achieve success rates comparable to white-box attacks, even when private models have been trained on proprietary data. Furthermore, our approach demonstrates strong transferability to other models, including LLaMA3 and ChatGPT. These findings highlight the urgent need for more robust defenses when fine-tuning open-source LLMs.

