# OpenReview forum: "Adversarial Attacks on Fine-tuned LLMs"
_ICLR.cc/2025/Conference — ICLR 2025 Conference Withdrawn Submission_

### Official Review · Reviewer_MiGQ · 2024-11-02

**Soundness:** 1
**Presentation:** 2
**Contribution:** 2
**Rating:** 3
**Confidence:** 5

**Summary:**

This paper proposes a new technique for generating suffix based jailbreak attacks against privately fine-tuned versions of open source LLMs. The threat model assumes that the adversary has white box access to the open source LLM and soft label (i.e. log probs) black box access to the fine-tuned version. The authors propose a iterative procedure where each step consists of two stages - (1) the adversarial suffix is first optimized to align the output distribution of the open and the fine-tuned LLM and then, (2) the resultant suffix is then optimized to jailbreak the open LLM. The authors evaluate their attack on Llama2-7b as the open model and Vicuna-7b as the fine-tuned version. They also compare against baselines where white box attacks generated against the open LLM are transferred to the fine-tuned version. The proposed attack achieves a higher attack success rate compared to the considered baseline. Overall, the paper considers a commonly used setting and provides a way to leverage the open source LLM while attacking it’s fine-tuned version.

**Strengths:**

Overall, this paper considers a realistic setting where open source LLMs are privately fine-tuned and deployed in a black box setting. Existing attacks only attack model individually, whereas, the proposed attack leverages the open LLM to incorporate additional information about the fine-tuned version. The 2-stage optimization procedure iteratively performs local (around the harmful input query) alignment of the open LLM with the fine-tuned version which increases the transferability of the generated jailbreak. Performing this alignment in each step of the attack iteration should improve transferability as compared to performing the alignment only once at the beginning of the attack.

**Weaknesses:**

The evaluation section of this paper leaves much to be desired. It is unclear whether using two-stage optimization provides any empirical benefit over directly attacking the black box fine-tuned LLM. The major weaknesses are as below:

- The paper considers a setting where the adversary has black box access to the fine-tuned model. It discusses one way to attack in the black box setting i.e. transferability, but it entirely misses mentioning anything about the the second way i.e. query based attacks which constitutes a rich literature of black box jailbreak attacks against LLMs [1,2,3,4]. To demonstrate that the proposed is actually useful, the authors need to evaluate against query based black box attacks.
- The authors only consider the setting of Llama2 → Vicuna. Evaluating only this setting is not enough to demonstrate that generalizability of the attack. Further, it fails to provide any insight on how does the attack success rate depend on the type of fine-tuning (standard or parameter efficient), which is an important part of the considered threat model. For example, guanaco is an openly available model which is QLoRA fine-tuned version of the Llama models.
- The evaluation uses a string matching based judge which can lead to false positives (since even without a negative prefix, the model response can still fail to successfully respond to the harmful query). Similar to recent work, the authors should instead use LLM-as-a-Judge to evaluate the success of the attack (there are open source judges as well) [5,6].

[1] Andriushchenko, Maksym, Francesco Croce, and Nicolas Flammarion. "Jailbreaking leading safety-aligned llms with simple adaptive attacks." arXiv preprint arXiv:2404.02151 (2024).

[2] Sitawarin, Chawin, et al. "Pal: Proxy-guided black-box attack on large language models." arXiv preprint arXiv:2402.09674 (2024).

[3] Hayase, Jonathan, et al. "Query-based adversarial prompt generation." arXiv preprint arXiv:2402.12329 (2024).

[4] Mehrotra, Anay, et al. "Tree of attacks: Jailbreaking black-box llms automatically." arXiv preprint arXiv:2312.02119 (2023).

[5] Mazeika, Mantas, et al. "Harmbench: A standardized evaluation framework for automated red teaming and robust refusal." arXiv preprint arXiv:2402.04249 (2024).

[6] Chao, Patrick, et al. "Jailbreakbench: An open robustness benchmark for jailbreaking large language models." arXiv preprint arXiv:2404.01318 (2024).

**Questions:**

1. In Section 5, the authors state that - “*Our framework assumes that the target model is only slightly fine-tuned from the original model. However, there may be a drop in ASR when the fine-tuned model significantly differs from the original. In such cases, as discussed in the experiments, our framework can still generate suffixes with high transferability.*” It is unclear what part of the evaluation supports this statement, since GPT3.5 etc are not fine-tuned from either Llama or Vicuna.
2. The plot in Figure 2 shows the loss values for the baseline and the proposed attack. Why does the proposed attack have a lower loss value at the beginning of the attack?
3. According to Table 1, the ASR for the proposed attack on Llama in the harmful behaviour case is 49 when it is the original model (white box setting) but 54 when it is the target model (black box setting). How is this possible ?

---

### Official Review · Reviewer_F7tU · 2024-11-03

**Soundness:** 2
**Presentation:** 2
**Contribution:** 2
**Rating:** 5
**Confidence:** 4

**Summary:**

The authors propose a setting for jailbreaking attacks, which targets private LLMs with inaccessible parameters and unknown fine-tuning data derived from public, open-weight LLMs. Additionally, the authors propose a method for obtaining suffixes for jailbreaking the private LLM, by optimizing suffixes on the public LLM. The authors claim that performance is comparable to attacks that incorporate information about the target LLM's parameters.

**Strengths:**

- The results are surprisingly strong for the target LLM in Table 1
- Table 2 suggests that transferability of the authors' method is significantly stronger than GCG
- The preliminaries are clear and the notation is rigorous

**Weaknesses:**

- The justification for the setting the authors propose is unclear. If one has access to the public base LLM weights, an attacker can simply fine-tune the model on harmful data to achieve the desired outputs.
- Prior work [1] has pointed out that benign fine-tuning already causes harmful ASR to increase by default, which seems likely to account for some of the ASR increase in this threat model
- There is a significant ASR increase in the black-box models in Table 2, though it doesn't seem obvious that the method should be significantly more successful than GCG in this regime, and the authors provide very limited commentary on this.
- The authors don't include evaluations against highly relevant defenses, such as RPO [2]
- The presentation needs work overall, particularly Figures 2 and 3

[1] Qi, X., Zeng, Y., Xie, T., Chen, P. Y., Jia, R., Mittal, P., & Henderson, P. (2023). Fine-tuning aligned language models compromises safety, even when users do not intend to!

[2] Zhou, A., Li, B., & Wang, H. (2024). Robust prompt optimization for defending language models against jailbreaking attacks.

**Questions:**

1) Can the authors justify why the ASR compared to GCG is much higher in Table 2?
2) Can the authors clarify the technical novelty of their method compared to GCG?

---

### Official Review · Reviewer_6FHU · 2024-11-03

**Soundness:** 2
**Presentation:** 2
**Contribution:** 2
**Rating:** 3
**Confidence:** 4

**Summary:**

This work develops an approach that generates adversarial examples using open-source models and fine-tunes them to target private, black-box models. The approach first searches for synthetic prompt suffices that align the open-source model generation with the private model generation. Then, standard attack methods are performed using the open-source model, and the synthetic prompt suffices. The open-source model combined with the suffices is considered as a proxy of the private fine-tuned model.

**Strengths:**

1. Using prompt suffixes to convert open-source models to a proxy of a private model is novel.

**Weaknesses:**

1. The motivation for optimizing the prompt is unclear. There are several ways to "steal" a private model or directly launch an attack without acquiring a proxy model of the private model. What are the limitations of existing approaches, and what is the key advantage of the prompt-based approach? Is it more affordable than LoRA?

2. The proposed approach underperforms against competitors such as PAIR. Although PAIR is starred in Table 2, I do not see a corresponding footnote or explanation.

3. The authors claim a novel setting that does not expect the generated attacks to succeed in targeting the public base LLM. However, the motivating scenario is not grounded in the paper.

4. The proposed attack has a 90% attack success rate on the public base LLM, according to Table 2, which contradicts the claim on reduced success rate on the public base LLM.

**Questions:**

1. Could the authors explain the difference between the proposed approach and PAIR? Section 4.2 mentioned that "PAIR achieves higher ASR than our method as it can query the downstream models to generate attacks. However, considering our focus on transferability evaluation, our performance approaches are achievable by querying black-box models.". How does the "query" in PAIR differ from yours?

2. Could the authors provide a compelling use case when transferring an attack back to the open-source model is prohibitive? Who will be the primary user of this method?

---

### Official Review · Reviewer_4Drs · 2024-11-04

**Soundness:** 2
**Presentation:** 2
**Contribution:** 1
**Rating:** 3
**Confidence:** 4

**Summary:**

This paper proposes a method to attack private, black-box LLMs that have been finetuned using open-source models on proprietary data. Black-box attacks are a well studied problem in literature. The only additional relaxation in this case is that the base model is known completely, and can be exploited to improve attack success rates on finetuned black-box models.

**Strengths:**

The paper addresses an important problem, with a new approach.

**Weaknesses:**

1. The threat model makes omissions regarding access to the target model; namely, the probability distributions of its outputs.
2. It is not clear what the "local data pairs" refer to in line 230. This makes it very hard to appreciate the optimization objective in eq (6). There is no justification given as to why aligning the model on arbitrary data pairs, would in any way be sufficient to find an adversarial suffix for the malicious query.
3. The justification for the approximation in eq.(5) does not hold for the chosen finetuned/base model pair. Vicuna is a finetuned version of Llama-2 that utilizes FSDP and flash attention to reduce memory overhead. It does not use a PEFT method that satisfies the assumption where most parameters of the base model are being frozen (https://github.com/lm-sys/FastChat/blob/main/fastchat/train/train.py). This cannot be used to explain the improvement in performance. This assumption also might not hold in instances where the finetuned model also gets extra alignment training, so it must be tested with more recent and robust finetuned/base model pairs. In this case, Vicuna is well known to be badly aligned (and can be seen as much from the results of the paper).
4. Vague comparisons in Table 2: In this comparison, the proposed method is optimizing over the original model, aside from their approximation of the target model. GCG only has access to the target model (it’s performance improves on this benchmark when optimized with other models: Pg. 13 https://arxiv.org/pdf/2307.15043)  and it’s not stated how many iterations does PAIR run compared to the other methods.
5. Authors haven't clearly explained the drop in target ASR of their attack for Vicuna->Llama as compared to the original model which is not case for other setting in Table 1.
6. Lack of models used in experiments. There are several recent and more robust finetuned/base model pairs available.
7. The judge used in the experiments is not appropriate. There can be many false positives, where the model is actually refusing to answer, but the refusal suffixes do not catch. For example, Vicuna can be very easily overfitted to the target string and not have anything after it.

**Questions:**

1. Could the authors provide more justification and intuition behind their approach?
2. Could the authors justify the relaxations made in the setup?
3. Could the authors provide more comparisons with other grey-box methods?
4. Lastly, could the authors clarify the setup used for PAIR and GCG in Table 2?

---

### Note · Authors · 2024-11-25

I have read and agree with the venue's withdrawal policy on behalf of myself and my co-authors.